# Skin Carotenoid Levels Are Associated with Demographic Factors, Body Size, and Fruit and Vegetable Intake in the Japanese Population

**DOI:** 10.3390/nu16132133

**Published:** 2024-07-04

**Authors:** Emiko Okada, Kayo Kurotani, Hidemi Takimoto

**Affiliations:** 1The Health Care Science Institute, 3-2-12 Akasaka, Minato-ku 107-0052, Tokyo, Japan; 2Department of Nutritional Epidemiology and Shokuiku, National Institute of Biomedical Innovation, Health and Nutrition, Kento Innovation Park, NK Building, 3-17 Senrioka Shinmachi, Settsu-shi 566-0002, Osaka, Japan; 3Department of Health Sciences, Showa Women’s University, 1-7-57 Taishido, Setagaya-ku 154-8533, Tokyo, Japan

**Keywords:** skin carotenoid, fruit and vegetable intake, demographic factors, body size, Japanese

## Abstract

This study aimed to determine the association between demographic factors, body size, and fruit and vegetable intake in the general population, focusing on individuals with both low and high skin carotenoid levels. This cross-sectional study was conducted during the 14th National Convention on the Promotion of Food and Nutrition Education (2019) in Yamanashi, Japan (a rural area) and the Open House 2019 at the National Institute of Biomedical Innovation, Health, and Nutrition in Tokyo, Japan (an urban area). Skin carotenoid measurements were conducted, and the participants were asked to fill out a self-administered questionnaire. The study population consisted of 492 Japanese individuals aged ≥16 years. The odds ratios (ORs) for low skin carotenoid levels were elevated in males, those who were overweight, and those who almost never consumed or consumed only one vegetable dish/day. Conversely, the ORs were lower in those living in Yamanashi, aged 30–39 and ≥70 years, and those who consumed fruit ≥1 time/week. For high skin carotenoid levels, the ORs were higher among those aged ≥70 years, living in Yamanashi, and those who consumed fruit ≥1 time/day or ≥5 vegetable dishes/day. Demographic factors, body size, and habitual fruit and vegetable intake may serve as indicators of skin carotenoid levels.

## 1. Introduction

The consumption of fruits and vegetables has beneficial effects on the prevention of non-communicable diseases (NCDs), such as diabetes [1], hypertension [2], cardiovascular disease, cancer, and all-cause mortality [3]. The World Health Organization recommends a daily intake of at least 400 g of fruits and vegetables for a healthy diet [4]. In “Health Japan 21 (the second term)”, a national health promotion measure that sets numerical targets in various fields including nutrition and diet, the targets are set to reduce the percentage of people whose fruit intake is less than 100 g to 30% and to increase vegetable intake to 350 g or more per day [5]. However, the percentage of Japanese people aged ≥20 years who consume less than 100 g of fruit is 61.6% and their vegetable consumption is 280.5 g (excluding potatoes) [6], posing a significant challenge due to these low intakes. Adolescents aged 15–19 consume an average of 243.4 g of vegetables [6], underscoring the need to cultivate healthy vegetable consumption habits from a young age.

Recently, methods using resonance Raman spectroscopy and reflection spectroscopy to detect carotenoids in human skin have gained attention as useful parameters for assessing vegetable and fruit intake [7]. Carotenoids, including α-carotene, β-carotene, β-cryptoxanthin, lycopene, lutein, astaxanthin, and zeaxanthins, are primarily found in dark orange, red, and green fruits and vegetables, offering antioxidant properties [8]. Nutrition education programs have been implemented for American parents and children [9], Mexican heritage families with children [10], Japanese workers [11], and Japanese elementary and junior high school students [12], aiming to improve diet by assessing skin carotenoid levels. The measurement of skin carotenoid levels is a non-invasive and objective method, minimising the burden on both participants and investigators [7]. Multiple studies have reported positive correlations between skin carotenoid levels and blood biomarkers (serum/plasma carotenoid concentrations) in different ethnic groups, such as non-Hispanic black/African American, non-Hispanic white, Hispanic/Latino, and Asian people [7,13,14]. Skin carotenoid levels have been associated with fruit and vegetable intake as assessed through a self-reported food frequency questionnaire survey [15,16] or 24-h dietary recall [17]; therefore, skin carotenoid measurements may serve as surrogates for fruit and vegetable intake in epidemiological studies. In Japanese workers, skin carotenoid levels are reportedly associated with the self-reported number of vegetable dishes consumed (70 g of vegetables per dish) [18]. A review of factors associated with skin carotenoid levels showed that sex, BMI, race/ethnicity, smoking status, carotenoid supplement intake, and recent sun exposure were associated with skin carotenoid levels [19]. However, information regarding the characteristics of study participants with low/high skin carotenoid levels in that population, including fruit and vegetable intake, is limited.

Therefore, this study aimed to examine individuals with low and high skin carotenoid levels, determine the population characteristics, and investigate the association between skin carotenoid levels, sex, age, area, body size, and self-reported fruit and vegetable intake. Identifying the characteristics of populations with low and high skin carotenoid levels is valuable for identifying target populations for future health and nutrition education.

## 2. Materials and Methods

### 2.1. Study Population

This cross-sectional study was conducted at an exhibition booth at the institution during the 14th National Convention on the Promotion of *Shokuiku* (Food and Nutrition Education) in Kofu, Yamanashi, Japan (a rural area) from 29 to 30 June 2019, and at the Open House event at the National Institute of Biomedical Innovation, Health, and Nutrition in Tokyo, Japan (an urban area) on 5 October 2019. This study involved Japanese individuals aged ≥16 years who were able to give their own consent to participate. In Japan, an adult is defined as being aged ≥18 years. The Ethical Guidelines for Medical and Health Research Involving Human Subjects stipulate that a study may be conducted with minors aged 16–17 years who have provided informed consent if they are deemed to have sufficient capacity to make informed decisions regarding the study. The National Convention on the Promotion of *Shokuiku* is an annual nationwide event co-sponsored by national and local governments. It serves as a core event during the “*Shokuiku* Month” in June, focusing on the intensive and effective promotion of nutrition education campaigns [20]. The Open House event was organised to present the institute’s daily research activities and findings to the general public. All visitors to both events participated voluntarily and were recruited to participate in the study when visiting the exhibition booths. Skin carotenoid levels were measured, and a self-administered questionnaire was conducted among visitors to either event who agreed to participate in this study. A total of 367 and 180 people participated in the Yamanashi and Tokyo events, respectively. Among the 547 participants, those whose skin carotenoid levels were not measured, those who did not complete the questionnaire survey, and those with missing information on age, body mass index (BMI), number of vegetable dishes, and frequency of intake of fruits, green and yellow vegetables, and light-coloured vegetables were excluded. Thus, 492 participants were included in the analysis. This study was conducted in accordance with the Declaration of Helsinki and the Ethical Guidelines for Medical and Health Research Involving Human Subjects. This study was approved by the Ethics Committee of the National Institute of Biomedical Innovation, Health, and Nutrition (protocol code: KENEI 107, date of approval: 20 June 2019). Written informed consent was obtained from all participants.

### 2.2. Measurement of Skin Carotenoid Levels

Skin carotenoid levels were measured using the Veggie Meter^®^ (Logevity Link Japan Corporation, Kanagawa, Japan). The Veggie Meter^®^ used in this study is the same device that has been used in studies conducted in other countries [7,12,14]. The Veggie Meter^®^ has been validated for use with adults and older adolescents to determine carotenoid levels in the skin [7]. The operational principles of this device are detailed in the literature [21]. Briefly, the Veggie Meter^®^ provides objective information about the concentration of carotenoid pigments in human skin via pressure-mediated reflection spectroscopy. Daily calibration using the supplied dark and white reference materials was performed before measurements were taken. All participants used disposable towels to disinfect their fingers. Subsequently, the middle finger of the left hand was inserted into the finger cradle of the device and was pressed against the convex contact lens surface using a spring-loaded lid. The averaging mode of the device was employed, recording the average score from three consecutive measurements as the participant’s skin carotenoid score. The skin carotenoid score is expressed as a value from 0 to 1200 and is unitless.

### 2.3. Dietary Survey

A self-administered questionnaire was used to gauge the number of vegetable dishes consumed, fruit intake, and frequency of food intake. In Japan, the amount of vegetables contained in one small bowl or other serving is about 70 g, and efforts to consume at least five dishes (servings) of vegetables per day are widely practiced [22,23]. One serving of a vegetable was also defined in this study as the equivalent of 70 g of vegetables, and the number of vegetable dishes was evaluated across six categories: almost never, 1, 2, 3, 4, and ≥5 dishes/day. Food intake frequency was assessed using a modified version of the 2013 National Health and Nutrition Survey questionnaire in Japan [24]. Vegetable and fruit juice, as well as milk and dairy products were added to the list of foods used in the 2013 National Health and Nutrition Survey [24]. Specifically, 14 food items were included: rice, bread, noodles, meat, fish and shellfish, eggs, soybeans and soybean products, green and yellow vegetables, light-coloured vegetables, vegetable and tomato juice, green juice with kale, fruit, 100% fruit juice, and milk and dairy products. These fourteen food items were assessed using five frequency categories: ≥1 time/day, 4–6 times/week, 2–3 times/week, 1 time/week, and <1 time/week. Additionally, information on dietary supplement use was collected, and the responses were categorised into never, rarely, sometimes, and every day.

### 2.4. Survey on Demographics and Behavioural Factors

Data on demographic characteristics and behavioural factors were gathered prior to measuring skin carotenoid levels, using the same self-administered questionnaire employed in the dietary survey. The gathered information included sex, age, height, weight, smoking status, passive smoking status, alcohol consumption status, physical activity level, and sleep quality. BMI was calculated by dividing weight (kg) by height in meters squared (m) (kg/m^2^).

### 2.5. Statistical Analysis

To identify participants with low or high skin carotenoid levels in this population, we defined low skin carotenoid levels as below the 25th percentile and high skin carotenoid levels as above the 75th percentile. The participants were initially divided into quartiles based on their skin carotenoid scores (first: <317, second: ≥317–<381, third: ≥381–<470, and fourth: ≥470). They were subsequently reclassified into three groups: low (below the first quartile), medium (above the first quartile but below the fourth quartile), and high (above the fourth quartile). A chi-square test was used to compare the characteristics of the participants and dietary intake between the groups categorised by skin carotenoid levels. A multivariate logistic regression analysis was conducted to assess the associations between low and high skin carotenoid levels and demographic factors, body size, and fruit and vegetable intake. The multivariate models were adjusted for sex, age group (16–19, 20–29, 30–39, 40–49, 50–59, 60–69, or ≥70 years), area (Yamanashi (a rural area) or Tokyo (an urban area)), BMI (<18.5 (underweight), 18.5–24.9 (normal weight), or ≥25.0 kg/m^2^ (overweight)), and the number of vegetable dishes consumed (almost never, 1, 2, 3, 4, or ≥5 dishes/day) or the frequency of fruit intake (≤1 time/week, 2–3 times/week, 4–6 times/week, or ≥1 time/day). Considering that vegetable dishes include both green, yellow, and light-coloured vegetables, and noting the observed association between the number of vegetable dishes and their intake frequency (Appendix A), the number of vegetable dishes consumed was incorporated into the model. All statistical analyses were performed using the SAS statistical package for Windows (version 9.4; SAS Institute Inc., Cary, NC, USA). A *p*-value of <0.05 was considered significant.

## 3. Results

The median skin carotenoid level was 381. The characteristics of the participants according to their skin carotenoid levels are shown in Table 1. The low skin carotenoid level group had a higher proportion of males, younger individuals, people residing in Tokyo, and overweight participants compared with the high skin carotenoid level group.

Dietary intake categorised by skin carotenoid levels is presented in Table 2. Participants with high skin carotenoid levels had higher consumption of vegetable dishes, less frequent consumption of noodles, and greater intake of meats, fish and shellfish, eggs, soybeans and soybean products, green and yellow vegetables, light-coloured vegetables, fruits, and milk and dairy products.

The results of the multivariate logistic regression analysis, indicating the odds ratios (ORs) for low and high skin carotenoid levels in association with the demographic factors, body size, and the number of vegetable dishes consumed, are presented in Table 3. The ORs for low skin carotenoid levels were higher in males than in females [OR: 2.82, 95% confidence interval (CI): 1.67–4.77], overweight individuals compared to normal weight individuals (OR: 5.02, 95% CI: 2.51–10.06), and those consuming less than two vegetable dishes per day compared to those consuming two dishes (almost never = OR: 2.78, 95% CI: 1.02–7.59; OR: 2.49, one vegetable dish per day = 95% CI: 1.39–4.47). The ORs were lower in those aged 30–39 years and ≥70 years compared with those aged 50–59 years (30–39 years old = OR: 0.36, 95% CI: 0.16–0.83; ≥70 years old = OR: 0.15, 95% CI: 0.04–0.58). Elevated ORs for high skin carotenoid levels were observed among participants aged ≥70 years, residing in Yamanashi, and consuming ≥5 vegetable dishes/day compared with the reference group.

The results of the multivariate logistic regression analysis, which examined the associations between demographic factors, body size, frequency of fruit intake, and low and high skin carotenoid levels, are provided in Table 4. The demographic factors and body size yielded results consistent with those presented in Table 3. The group that consumed fruits ≥1 time/day had lower ORs for low skin carotenoid levels (OR: 0.22, 95% CI: 0.09–0.55) and higher ORs for high skin carotenoid levels (OR: 3.39, 95% CI: 1.66–6.96) compared with the group that consumed fruit 4–6 times/week.

## 4. Discussion

We examined individuals with low or high skin carotenoid levels in the general population to explore associations between demographic factors, body size, and self-reported fruit and vegetable intake. In the multivariate model, males and overweight individuals were associated with low skin carotenoid levels, whereas older age, rural event attendance, and higher consumption of vegetables or fruits were associated with high skin carotenoid levels.

Numerous studies have demonstrated sex differences in skin carotenoid levels, typically indicating higher levels in females compared to males [13,25,26,27]. The present study similarly observed low skin carotenoid levels in males. A previous study suggested that plasma carotenoid concentrations in males and females are influenced by high-density lipoprotein (HDL) cholesterol and body weight [28]. Although our study included body size in the multivariate model, plasma lipid levels were not investigated. Moreover, similar to some studies [15,25,26], our findings indicated higher skin carotenoid levels in older participants. However, the existing literature presents conflicting results, as some studies have found no significant associations [13,14]. Additionally, the participants in their 30s exhibited lower skin carotenoid levels. Despite adjusting for fruit and vegetable intake in the multivariate model, the precise mechanisms underlying the effects of age on carotenoid levels remain unclear.

Event attendees in the rural area of Yamanashi exhibited higher skin carotenoid levels than those in the urban area of Tokyo. A previous study showed that urban shoppers in New York City had higher skin carotenoid levels than those in rural North Carolina [29]. Their findings contrast with the results of our study and may be attributed to variations in country-specific characteristics and access to fruits and vegetables. According to the National Health and Nutrition Survey in Japan, daily vegetable intake was 318 g for males and 300 g for females in Yamanashi, and 275 g for males and 277 g for females in Tokyo [30]. However, in our study, adjustment for the number of vegetable dishes consumed in the multivariate model revealed regional differences. Furthermore, considering that the events in Yamanashi and Tokyo were held in June and October, respectively, it was not feasible to ascertain whether the discrepancies in skin carotenoid levels could be attributed to regional or seasonal differences. A previous study reported that skin carotenoid scores measured in autumn were higher than those measured in winter [16], although another study reported little seasonal differences [31]. Further studies should be undertaken to compare and examine skin carotenoid levels in rural and urban residences in each country, taking into consideration the potential seasonal differences.

In line with studies investigating skin carotenoid levels and physical characteristics [13,15], our findings suggest an association between low skin carotenoid levels and being overweight. Recent studies, including meta-analyses, have indicated that low serum carotenoid levels are risk factors for overweight and obesity [32,33]. Furthermore, oral carotenoid administration has demonstrated promising effects on anthropometric and lipid metabolic parameters in overweight or obese individuals by decreasing body weight and BMI and increasing HDL cholesterol levels [33]. This evidence supports the results of this study. Skin carotenoid levels could serve as a non-invasive method for identifying individuals at risk of dyslipidaemia and a marker for improvements in blood lipid levels and body weight. Further studies are needed to elucidate this relationship.

The association between self-reported high fruit and vegetable intake and high skin carotenoid levels is consistent with the results of numerous previous studies [15,16,29,34]. In another Japanese study, skin carotenoid levels increased as the self-reported number of vegetable dishes consumed per day increased [18]. Five servings of vegetable dishes are equivalent to 350 g of vegetables, corresponding to the recommended daily vegetable intake for adults aged ≥20 years under Health Japan 21 (the second term) [5]. Assessing vegetable intake based on the number of dishes offers a more convenient alternative to conducting a comprehensive dietary survey, making it a valuable tool in nutrition education and dietary assessment. Our findings suggest that low skin carotenoid levels can be determined by low vegetable/infrequent fruit intake and high vegetable/frequent fruit intake, regardless of demographic factors or body size. The scientific evidence provided by the current dietary guidelines in Japan, as indicated by the results of this study, appears to be valid [5]. Additionally, those who consumed a greater number of vegetable dishes and had more frequent fruit intake exhibited a higher percentage of meat, fish, soy, and dairy product consumption at least once daily (Appendix A). Skin carotenoid levels may serve as an indicator for dietary balance in addition to reflecting fruit and vegetable intake. The anticipated utility of skin carotenoid levels in health and nutrition education further underscores their potential.

This study’s primary strength lies in its focus on evaluating the general population of Japanese adults and older adolescents. Another notable strength is the analytical method used to categorise skin carotenoids into low and high levels, facilitating their assessment in everyday situations. However, this study has several limitations. First, the study population comprised those who attended events related to food and nutrition education and an Open House at the institute. The participants were likely interested in nutrition and health, which may have influenced the results. Caution is warranted when generalizing the results as this study included participants with high levels of health awareness. Second, skin carotenoid levels may be influenced by the intake of other foods, dietary supplements, and sun exposure [35]. We identified the effects of food and dietary supplement intake in a multivariate model that showed associations with skin carotenoid levels in univariate analyses. However, these intakes were not associated with skin carotenoid levels. Sun exposure was not investigated and requires further examination. Third, other devices have been used to measure skin carotenoid levels [36]. In this study, skin carotenoids were assessed using the Veggie Meter^®^, and measurements may vary when different devices are employed. Future considerations should include the standardisation of skin carotenoid measurements. Fourth, dietary surveys were conducted based on self-reported daily food intake frequency. In general, conducting a multiple-day survey using the dietary record method or the 24-h recall method can reduce within-individual variation and provide a more accurate estimate of habitual dietary intake [37]. Finally, although we examined habitual dietary intake, including the number of vegetable dishes consumed and the frequency of food intake, quantitative dietary surveys were not conducted. Therefore, the quantitative relationship between nutrient and food intake and skin carotenoid levels could not be established. However, defining one serving of a vegetable dish as 70 g of green, yellow, and light-coloured vegetables combined allows for a rough estimation of the amount of vegetables consumed. Future epidemiological studies should explore whether skin carotenoid levels, a parameter of fruit and vegetable intake, are associated with NCD risk.

## 5. Conclusions

This study revealed that in the Japanese general population, being male and being overweight is associated with low skin carotenoid levels. On the contrary, older age, rural event attendance, and higher consumption of vegetables and fruits are associated with high skin carotenoid levels. These findings may benefit health and nutrition education in Japan.

## Figures and Tables

**Table 1 nutrients-16-02133-t001:** Characteristics of the participants according to their skin carotenoid level category (*n* = 492).

		Skin Carotenoid Levels						
		Low (<317), *n* = 122		Medium (317–<470), *n* = 247		High (≥470), *n* = 123		
		*n*	(%)	*n*	(%)	*n*	(%)	*p*-Value
Sex	Males	49	(40.2)	48	(19.4)	21	(17.1)	<0.001
	Females	73	(59.8)	199	(80.6)	102	(82.9)	
Age (years)	16–19	5	(4.1)	13	(5.3)	4	(3.3)	<0.001
	20–29	47	(38.5)	66	(26.7)	16	(13.0)	
	30–39	14	(11.5)	39	(15.8)	22	(17.9)	
	40–49	19	(15.6)	41	(16.6)	18	(14.6)	
	50–59	28	(23.0)	45	(18.2)	24	(19.5)	
	60–69	6	(4.9)	23	(9.3)	18	(14.6)	
	≥70	3	(2.5)	20	(8.1)	21	(17.1)	
Area	Yamanashi	62	(50.8)	155	(62.8)	98	(79.7)	<0.001
	Tokyo	60	(49.2)	92	(37.3)	25	(20.3)	
BMI (kg/m2)	<18.5	15	(12.3)	28	(11.3)	23	(18.7)	<0.001
	18.5–24.9	78	(63.9)	202	(81.8)	94	(76.4)	
	≥25	29	(23.8)	17	(6.9)	6	(4.9)	
Smoking status	Never smoker	105	(86.1)	222	(89.9)	118	(95.9)	0.128
	Former smoker	8	(6.6)	11	(4.5)	2	(1.6)	
	Current smoker	9	(7.4)	13	(5.3)	3	(2.4)	
	Unknown	0	(0.0)	1	(0.4)	0	(0.0)	
Passive smoking status	Never	45	(36.9)	103	(41.7)	62	(50.4)	0.068
	1 time/month	24	(19.7)	55	(22.3)	29	(23.6)	
	1 time/week	15	(12.3)	35	(14.2)	13	(10.6)	
	Few times/week	22	(18.0)	28	(11.3)	6	(4.9)	
	Almost every day	15	(12.3)	23	(9.3)	10	(8.1)	
	Unknown	1	(0.8)	3	(1.2)	3	(2.4)	
Alcohol drinking status	Non-drinker	40	(32.8)	93	(37.7)	61	(49.6)	0.056
	1–3 days/month	35	(28.7)	74	(30.0)	27	(22.0)	
	1–2 days/week	17	(13.9)	37	(15.0)	11	(8.9)	
	3–4 days/week	7	(5.7)	14	(5.7)	6	(4.9)	
	5–6 days/week	5	(4.1)	13	(5.3)	8	(6.5)	
	Every day	18	(14.8)	15	(6.1)	8	(6.5)	
	Unknown	0	(0.0)	1	(0.4)	2	(1.6)	
Physical activity								
Muscular work or strenuous sports (hours)	None	72	(59.0)	150	(60.7)	57	(46.3)	0.048
	<1	30	(24.6)	58	(23.5)	46	(37.4)	
	≥1	20	(16.4)	37	(15.0)	18	(14.6)	
	Unknown	0	(0.0)	2	(0.8)	2	(1.6)	
Sedentary (hours)	>3	12	(9.8)	40	(16.2)	19	(15.5)	0.387
	3–<8	88	(72.1)	161	(65.2)	76	(61.8)	
	≥8	22	(18.0)	45	(18.2)	27	(22.0)	
	Unknown	0	(0.0)	1	(0.4)	1	(0.8)	
Walking or standing (hours)	<1	21	(17.2)	39	(15.8)	28	(22.8)	0.114
	1–<3	68	(55.7)	128	(51.8)	50	(40.7)	
	≥3	32	(26.2)	77	(31.2)	45	(36.6)	
	Unknown	1	(0.8)	3	(1.2)	0	(0.0)	
Sleep quality	Enough	21	(17.2)	60	(24.3)	25	(20.3)	0.447
	Fair enough	63	(51.6)	127	(51.4)	65	(52.9)	
	Not much or not at all	38	(31.2)	59	(23.9)	31	(25.2)	
	Unknown	0	(0.0)	1	(0.4)	2	(1.6)	

A chi-square test was used. Missing values were excluded in the analysis of each item.

**Table 2 nutrients-16-02133-t002:** Dietary intake according to skin carotenoid level category (*n* = 492).

		Skin Carotenoid Levels						
		Low (<317), *n* = 122		Medium (317–<470), *n* = 247		High (≥470), *n* = 123		
		*n*	(%)	*n*	(%)	*n*	(%)	*p*-Value
Number of vegetable dishes (dish/day)	Almost never	12	(9.8)	10	(4.1)	3	(2.4)	<0.001
	1	56	(45.9)	65	(26.3)	20	(16.3)	
	2	29	(23.8)	73	(29.6)	37	(30.1)	
	3	14	(11.5)	54	(21.9)	27	(22.0)	
	4	9	(7.4)	23	(9.3)	15	(12.2)	
	≥5	2	(1.6)	22	(8.9)	21	(17.1)	
Frequency of food group intake in the last month								
Rice	≤2–3 times/week	7	(5.7)	15	(6.1)	5	(4.1)	0.257
	4–6 times/week	23	(18.9)	29	(11.7)	13	(10.6)	
	≥1 time/day	92	(75.4)	203	(82.2)	104	(84.6)	
	Unknown	0	(0.0)	0	(0.0)	1	(0.8)	
Breads	≤1 time/week	38	(31.2)	78	(31.6)	41	(33.3)	0.604
	2–3 times/week	38	(31.2)	62	(25.1)	24	(19.5)	
	4–6 times/week	15	(12.3)	34	(13.8)	18	(14.6)	
	≥1 time/day	29	(23.8)	67	(27.1)	37	(30.1)	
	Unknown	2	(1.6)	6	(2.4)	3	(2.4)	
Noodles	<1 time/week	26	(21.3)	51	(20.7)	39	(31.7)	<0.001
	1 time/week	27	(22.1)	81	(32.8)	47	(38.2)	
	2–3 times/week	48	(39.3)	69	(27.9)	18	(14.6)	
	≥4–6 times/week	19	(15.6)	38	(15.4)	13	(10.6)	
	Unknown	2	(1.6)	8	(3.2)	6	(4.9)	
Meats	≤1 time/week	10	(8.2)	9	(3.6)	8	(6.5)	0.024
	2–3 times/week	31	(25.4)	59	(23.9)	28	(22.8)	
	4–6 times/week	51	(41.8)	93	(37.7)	33	(26.8)	
	≥1 time/day	29	(23.8)	84	(34.0)	53	(43.1)	
	Unknown	1	(0.8)	2	(0.8)	1	(0.8)	
Fish and shellfish	≤1 time/week	41	(33.6)	53	(21.5)	21	(17.1)	<0.001
	2–3 times/week	56	(45.9)	102	(41.3)	40	(32.5)	
	4–6 times/week	18	(14.8)	57	(23.1)	29	(23.6)	
	≥1 time/day	6	(4.9)	33	(13.4)	33	(26.8)	
	Unknown	1	(0.8)	2	(0.8)	0	(0.0)	
Eggs	≤1 time/week	29	(23.8)	33	(13.4)	13	(10.6)	0.008
	2–3 times/week	42	(34.4)	82	(33.2)	36	(29.3)	
	4–6 times/week	27	(22.1)	68	(27.5)	29	(23.6)	
	≥1 time/day	23	(18.9)	63	(25.5)	45	(36.6)	
	Unknown	1	(0.8)	1	(0.4)	0	(0.0)	
Soybeans and soybean products	≤1 time/week	42	(34.4)	40	(16.2)	14	(11.4)	<0.001
	2–3 times/week	43	(35.3)	64	(25.9)	21	(17.1)	
	4–6 times/week	22	(18.0)	61	(24.7)	30	(24.4)	
	≥1 time/day	15	(12.3)	78	(31.6)	58	(47.2)	
	Unknown	0	(0.0)	4	(1.6)	0	(0.0)	
Green and yellow vegetables	≤1 time/week	21	(17.2)	20	(8.1)	6	(4.9)	<0.001
	2–3 times/week	32	(26.2)	47	(19.0)	12	(9.8)	
	4–6 times/week	40	(32.8)	69	(27.9)	22	(17.9)	
	≥1 time/day	29	(23.8)	111	(44.9)	83	(67.5)	
Light coloured vegetables	≤1 time/week	20	(16.4)	17	(6.9)	3	(2.4)	<0.001
	2–3 times/week	30	(24.6)	42	(17.0)	9	(7.3)	
	4–6 times/week	44	(36.1)	61	(24.7)	24	(19.5)	
	≥1 time/day	28	(23.0)	127	(51.4)	87	(70.7)	
Fruits	≤1 time/week	66	(54.1)	79	(32.0)	31	(25.2)	<0.001
	2–3 times/week	29	(23.8)	59	(23.9)	20	(16.3)	
	4–6 times/week	19	(15.6)	46	(18.6)	13	(10.6)	
	≥1 time/day	8	(6.6)	63	(25.5)	59	(48.0)	
Milk and dairy products	≤1 time/week	44	(36.1)	48	(19.4)	16	(13.0)	<0.001
	2–3 times/week	16	(13.1)	49	(19.8)	9	(7.3)	
	4–6 times/week	26	(21.3)	35	(14.2)	21	(17.1)	
	≥1 time/day	36	(29.5)	115	(46.6)	76	(61.8)	
	Unknown	0	(0.0)	0	(0.0)	1	(0.8)	
Dietary supplement use	Never	93	(76.2)	162	(65.6)	71	(57.7)	0.026
	Rarely	7	(5.7)	18	(7.3)	7	(5.7)	
	Sometimes	5	(4.1)	21	(8.5)	9	(7.3)	
	Every day	17	(13.9)	43	(17.4)	36	(29.3)	
	Unknown	0	(0.0)	3	(1.2)	0	(0.0)	

A chi-square test was used. Missing values were excluded in the analysis of each item.

**Table 3 nutrients-16-02133-t003:** Logistic regression analysis of the association between skin carotenoid levels and the number of vegetable dishes consumed (*n* = 492).

		Skin Carotenoid Levels							
		Low (<317)				High (≥470)			
		OR	(95% CI)		*p*-Value	OR	(95% CI)		*p*-Value
Sex	Males	2.82	(1.67,	4.77)	<0.001	0.66	(0.37,	1.17)	0.157
	Females	Ref.				Ref.			
Age (years)	16–19	0.52	(0.16,	1.71)	0.278	0.99	(0.28,	3.44)	0.986
	20–29	0.90	(0.46,	1.79)	0.772	0.64	(0.30,	1.35)	0.240
	30–39	0.36	(0.16,	0.83)	0.016	1.42	(0.69,	2.91)	0.338
	40–49	0.62	(0.29,	1.34)	0.227	0.89	(0.43,	1.85)	0.760
	50–59	Ref.				Ref.			
	60–69	0.37	(0.13,	1.05)	0.061	1.84	(0.84,	4.02)	0.129
	≥70	0.15	(0.04,	0.58)	0.006	2.81	(1.29,	6.13)	0.010
Area	Yamanashi	0.61	(0.36,	1.03)	0.066	2.06	(1.21,	3.53)	0.008
	Tokyo					Ref.			
BMI (kg/m^2^)	<18.5	1.50	(0.74,	3.04)	0.262	1.52	(0.83,	2.79)	0.179
	18.5–24.9	Ref.				Ref.			
	≥25	5.02	(2.51,	10.06)	<0.001	0.40	(0.16,	1.01)	0.052
Number of vegetable dishes consumed (dish/day)	Almost never	2.78	(1.02,	7.59)	0.047	0.51	(0.13,	1.98)	0.331
	1	2.49	(1.39,	4.47)	0.002	0.56	(0.30,	1.06)	0.075
	2	Ref.				Ref.			
	3	0.78	(0.37,	1.66)	0.522	0.96	(0.52,	1.79)	0.907
	4	0.84	(0.33,	2.09)	0.699	1.24	(0.58,	2.68)	0.576
	≥5	0.24	(0.05,	1.10)	0.067	2.13	(1.01,	4.46)	0.046

OR, odds ratio; CI, confidence interval; Ref., reference. The multivariate logistic regression model included the variables in the table.

**Table 4 nutrients-16-02133-t004:** Logistic regression analysis of the association between skin carotenoid levels and the frequency of fruit intake (*n* = 492).

		Skin Carotenoid Levels							
		Low (<317)				High (≥470)			
		OR	(95% CI)		*p*-Value	OR	(95% CI)		*p*-Value
Sex	Males	2.84	(1.67,	4.84)	<0.001	0.62	(0.35,	1.10)	0.101
	Females	Ref.				Ref.			
Age (years)	16–19	0.62	(0.19,	2.06)	0.438	0.95	(0.27,	3.27)	0.929
	20–29	0.93	(0.48,	1.82)	0.837	0.69	(0.32,	1.47)	0.334
	30–39	0.34	(0.15,	0.77)	0.010	1.57	(0.76,	3.23)	0.223
	40–49	0.63	(0.29,	1.35)	0.234	0.92	(0.44,	1.91)	0.813
	50–59	Ref.				Ref.			
	60–69	0.43	(0.15,	1.24)	0.119	1.56	(0.70,	3.49)	0.278
	≥70	0.21	(0.05,	0.81)	0.024	2.19	(0.98,	4.88)	0.055
Area	Yamanashi	0.56	(0.34,	0.94)	0.028	2.13	(1.24,	3.67)	0.006
	Tokyo	Ref.				Ref.			
BMI (kg/m^2^)	<18.5	1.66	(0.82,	3.33)	0.157	1.49	(0.81,	2.72)	0.200
	18.5–24.9	Ref.				Ref.			
	≥25	4.82	(2.42,	9.61)	<0.001	0.46	(0.18,	1.15)	0.097
Frequency of fruit intake	≤1 time/week	1.38	(0.72,	2.65)	0.335	1.26	(0.60,	2.64)	0.540
	2–3 times/week	1.04	(0.51,	2.12)	0.916	1.22	(0.55,	2.72)	0.622
	4–6 times/week	Ref.				Ref.			
	≥1 time/day	0.22	(0.09,	0.55)	0.001	3.39	(1.66,	6.96)	0.001

OR, odds ratio; CI, confidence interval; Ref., reference. The multivariate logistic regression model included the variables in the table.

## Data Availability

The data presented in this study are available on request from the corresponding author. The data are not publicly available due to privacy and ethical concerns.

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
