# Peer review of "Skin Carotenoid Levels Are Associated with Demographic Factors, Body Size, and Fruit and Vegetable Intake in the Japanese Population"

_nutrients, 2024, doi:10.3390/nu16132133_

Round 1

Reviewer 1 Report

Comments and Suggestions for Authors

Thank you for this careful and excellent study and manuscript! It reads very well and this reviewer has only a very few minor editorial comments. First, I want to thank you for the excellent Introduction. It sets the tone for the manuscript and fully provides the background needed for the reader.

Here are the few editorial comments:

line 38: replace the word "reaches" with "is"

line 102: change to read: "... used in studies in other countries" 

line 117: change to read: "One dish of a vegetable was also defined..."

lines 164-168: very nicely written! 

line 254: Please add a reference to the end of the sentence that reads:"... by the current dietary guidelines in Japan."

line 269: It appears that there is a repetition here that is not needed: Please delete the sentence: "Caution is warranted when generalizing the results". The previous sentence conveys the same information.

Again, congratulations for such a strong study and very well written manuscript!

Reviewer 2 Report

Comments and Suggestions for Authors

Thank you for submitting the manuscript "Skin Carotenoid Levels Were Associated with Demographic Factors, Body Size, and Fruit and Vegetable Intake in the Japanese Population" to Nutrients. The manuscript is well written and has relevant data on the relationship between carotenoid content in the Japanese population and the way they eat. The study protocol seems to have been well conducted, but I believe that some explanations need to be improved in the manuscript.

- How was the research conducted with children under 18? Please add an explanation of what is considered adulthood in Japan.

- Lines #138 and #141: why was this division established? Provide more information.

- Where was this list of foods used in the questionnaire taken from? Are foods typically commonly consumed in Japan? Add an explanation in M&M.

- Also add to M&M whether there is already validation for this equipment used to determine the concentration of carotenoids in the skin, it has already been validated in this population.

- Normally, to estimate food intake, it is accepted in the literature that an assessment is carried out on at least three non-consecutive days and an average is established. As this study was carried out with a self-report of just one day, I believe it is necessary to discuss this point as a limitation of the work.
